# Effects of Different Feeding Methods on the Structure, Metabolism, and Gas Production of Infant and Toddler Intestinal Flora and Their Mechanisms

**DOI:** 10.3390/nu14081568

**Published:** 2022-04-09

**Authors:** Xionge Pi, Hanju Hua, Qi Wu, Xiaorong Wang, Xin Wang, Jinjun Li

**Affiliations:** 1Institute of Plant Protection and Microbiology, Zhejiang Academy of Agricultural Sciences, Hangzhou 310021, China; pixionge@163.com; 2Department of Proctology, The First Affiliated Hospital, Zhejiang University School of Medicine, Hangzhou 310021, China; tomorrow97@163.com (H.H.); 1506143@zju.edu.cn (Q.W.); 3College of Animal Science, Shanxi Agricultural University, Taigu 030801, China; 13935472174@163.com; 4State Key Laboratory for Managing Biotic and Chemical Threats to the Quality and Safety of Agro-Products, Zhejiang Academy of Agricultural Sciences, Hangzhou 310021, China; xxww101@sina.com; 5Institute of Food Sciences, Zhejiang Academy of Agricultural Sciences, Hangzhou 310021, China

**Keywords:** gas production, fucosyllactose, prebiotics

## Abstract

In this study, we evaluated the effects of different feeding methods on the characteristics of intestinal flora and gas production in infants and toddlers by using an in vitro simulated intestinal microecology fermentation and organoid model. We found that the feeding method influences intestinal gas and fecal ammonia production in infants and toddlers. Supplementation with milk powder for infants in the late lactation period could promote the proliferation of beneficial bacteria, including *Bifidobacteria*. Intestinal flora gas production in a culture medium supplemented with fucosyllactose (2′-FL) was significantly lower than that in media containing other carbon sources. In conclusion, 2′-FL may reduce gas production in infant and toddler guts through two mechanisms: first, it cannot be used by harmful intestinal bacteria to produce gas; second, it can inhibit intestinal mucosa colonization by harmful bacteria by regulating the expression of intestinal epithelial pathogenic genes/signaling pathways, thus reducing the proliferation of gas-producing harmful bacteria in the gut.

## 1. Introduction

The human gut contains trillions of microorganisms. The structure and function of the microbial flora affect host health and are associated with the development of a wide range of diseases [1,2]. Although low numbers of bacteria in the amniotic fluid and placenta have been reported in recent years [3], the current consensus in the academic community is that bacteria begin to colonize and rapidly multiply in the gut at the moment of birth, which is when the intestinal micro-ecosystem is established [4]. In 2016, Matsuki et al. employed high-throughput sequencing to analyze the microflora structure of 12 normally delivered neonates during the first month of life and found significant individual differences in the structure of the neonatal intestinal flora. At the end of the first month, neonatal intestinal flora could be divided into three types: Bifidobacteria, Enterobacteriaceae, and *Staphylococcus aureus*. In fact, the structure of the neonatal intestinal flora during the first month involves a dynamic gradual evolution process from initial colonization with *S. aureus* to Enterobacteriaceae, finally progressing to Bifidobacteria. Although Bifidobacteria are the dominant bacteria in most neonatal guts, there is still a small proportion (20%) of neonates whose intestinal flora remains at the Enterobacteriaceae stage, and the Bifidobacteria population does not develop at the end of the first month [5]. The diversity of intestinal flora increases monthly after birth. The structure and diversity of the intestinal flora of children at 2 weeks of age approach that of adults, i.e., *Firmicutes* and *Bacteroidetes* are the dominant flora, and Bifidobacteria are relegated to secondary flora [6,7].

Compared with adults, the diet of infants is relatively simple and consists primarily of breast milk. Human milk contains hundreds of structurally different human milk oligosaccharides (HMOs) [8]. A normal liter of breast milk contains approximately 60.8 g of lactose, 40.5 g of lipids, 11.1 g of protein, and 12–24 g of oligosaccharides [9]. Due to the significant growth-stimulating effect of HMOs on Bifidobacteria, prebiotic products such as oligogalactose and oligofructose are widely added to infant formulas [10,11,12].

The microbial community in the gut matures, similar to the immune system of the infant [13,14]. Gut microbes play a significant role in the maturation of the host immune and digestive systems [15,16], and metabolic imbalance caused by dysbiosis can lead to the development of many diseases [17]. The metabolites of human gut microbes include short-chain fatty acids, medium- and long-chain fatty acids, gases, vitamins, organic amines, cresols, indoles, and other products that may be hazardous to health, such as neurotoxins, carcinogenic compounds, and immunotoxins [18,19]. Compared with the increasing attention paid to the effects of other metabolites on human health, relatively little attention has been paid to the health effects of gases on the host.

Neonatal abdominal distension or infantile abdominal distension is one of the most common and difficult symptoms to manage in clinical practice [20]. Neonatal abdominal distension is primarily gas distension and can be caused by various reasons, such as improper feeding, indigestion, abdominal infection, perforation, and necrotizing enterocolitis in newborns. The severity of abdominal distension and accompanying symptoms vary by cause. Although the physiopathology is mixed, dysbiosis caused by infection and antibiotic use is the leading cause of abdominal distension in neonates [21,22]. The recent discovery of *Clostridium neonatale*, the causative agent of neonatal necrotizing enterocolitis, which produces large amounts of gas after carbohydrate fermentation [23], further clarifies the role of infectious factors in abdominal distension in infants [22]. The structure of the intestinal flora of newborns varies greatly among individuals, and a significant proportion of normal infants do not have a predominantly *Bifidobacteria* intestinal flora [5]. *Bifidobacteria* are one of the very few genera of human intestinal flora that do not produce hydrogen gas [24], and the gut gas production of infants with a *Bifidobacteria*-dominant flora should be relatively low. Infants with other bacteria dominating their intestines are hypothesized to more likely suffer from gas production and abdominal distension.

Prebiotics, such as oligofructose and oligogalactose, benefit the host by selectively stimulating the growth and activity of one or a few beneficial intestinal bacterial species. Compared with breast milk, almost all formulas contain oligofructose and oligogalactose [25]. However, since prebiotics are oligosaccharide carbohydrates, it is unknown whether their addition to infant formula causes gas production by infant intestinal microorganisms, resulting in abdominal distension. This is not only an urgent issue for clinicians and parents but is also an important issue in intestinal microecology research.

Studies on gut bacterial gases are methodologically more difficult to perform than studies on other metabolites. Currently, the most effective approach is to measure hydrogen and methane levels following the consumption of indigestible carbohydrates by the subject [26,27]. However, this method has significant limitations. First, children under 3 years of age are too young to cooperate. Second, fasting for more than 12 h before the test is required, and the administration of large amounts of undigested oligosaccharides on the following day often exacerbates clinical symptoms in patients with diarrhea. Third, only one oligosaccharide can be measured per test, and comparative studies of multiple substrates cannot be performed simultaneously. Therefore, the in vitro human intestinal microecology simulation system combined with rapid gas assay analysis has become an effective experimental tool for studying intestinal microbial gas production.

Compared with the complex intestinal flora of adults, the structure of the intestinal flora and diet in infancy is relatively simple. However, there are still inter-individual differences [28], which result in differences in degradation, utilization, and gas production rates and gas production types from dietary carbohydrates such as oligosaccharides and polysaccharides. Since the medium in in vitro gut microbial simulation systems can be purposefully designed and adjusted, in vitro gut microbial simulation systems enable direct investigation and assessment of the effects of different prebiotics on the type of gut metabolites, especially gas production in infants. This is useful for comparative analysis of the association between specific food components and metabolic characteristics of the intestinal flora in healthy infants. Furthermore, such simulation systems help reveal the complex relationships between abdominal distension, flora structure, and dietary components and provide theoretical and practical guidance for formula development, especially for formula selection in infants with abdominal distension.

## 2. Materials and Methods

### 2.1. Collection and Treatment of Fecal Samples

Sixty fecal samples were collected from infants who were exclusively formula-fed or breastfed. Ten subjects aged ≤1, 1–3, and 3–6 months (growth stages 1, 2, and 3, respectively) were enrolled for each feeding method. All infants were healthy, and none had any illnesses or gastrointestinal problems. The study and operation procedures were in accordance with the ethical standards specified by the Medical Ethics Committee of the Child Health Care Hospital (2021CS036), and informed consent was obtained from the children’s parents.

Stool samples were collected at the Jianggan District Zhan Nong Kou Street Maternal and Child Health Care Hospital, Hangzhou, China, by scraping from diapers; samples were then placed in a plastic bag, kept at ambient temperature, and delivered to the laboratory in two batches. Stool processing began within 1 h of defecation. First, fresh fecal samples (0.8 g) were homogenized with 8 mL of 0.1 M anaerobic phosphate-buffered saline (PBS; pH 7.0) using an automatic fecal homogenizer (Halo Biotechnology Co. Ltd., Jiangsu, China) to prepare a 10% (*w*/*v*) fecal slurry. 

### 2.2. In Vitro Fermentation Method

Batch culture fermentation was conducted using the procedure described by Lei et al. The basic growth medium VI used contained the following: 4.5 g/L yeast extract, 3.0 g/L tryptone, 3.0 g/L peptone, 0.4 g/L bile salt No. 3, 0.8 g/L l-cysteine hydrochloride, 4.5 g/L NaCl, 2.5 g/L KCl, 0.45 g/L MgCl_2_.6H_2_O, 0.2 g/L CaCl_2_.6H_2_O, 0.4 g/L KH_2_PO_4_, 1.0 mL Tween 80, 1.0 mL resazurin, and 2.0 mL of a trace element solution. To assess the degradation and utilization of oligosaccharides by the infant fecal microbiome, 8.0 g each of FOS, GOS, lactose, or fucosyllactose (2′-FL) was added to the growth medium as the sole carbon sources, and 5 mL of test medium was dispensed into a 10-mL bottle under anaerobic conditions. Thereafter, 1-mL aliquots were removed at 0, 24, and 48 h for further analysis. The cultures were subsequently centrifuged, and the precipitates were collected and stored at 30 °C for DNA extraction.

### 2.3. DNA Extraction

Fecal samples were treated with a buffer and homogenization beads, mixed evenly, and heated at 70 °C for 15 min for lysis. This was followed by repeated centrifugation and supernatant transfer at 12,000 rpm for a total of five times. An elution buffer was then added, and the mixture was incubated at 56 °C for 5 min. After complete adsorption of the magnetic beads, the DNA solution was transferred to a new centrifuge tube and appropriately stored.

Bacterial genomic DNA was isolated from fermentation samples obtained at 0 and 48 h using a QIAamp DNA Stool Mini Kit according to the manufacturer’s instructions (Qiagen, Hilden, Germany). The concentration of extracted DNA was determined using a NanoDrop ND-2000 spectrophotometer (NanoDrop Technologies LLC, Wilmington, DE, USA), and its integrity and size were confirmed by agarose gel electrophoresis (1.0%) and stored at −20 °C.

### 2.4. Short-Chain Fatty Acid Analysis

A fecal sample of 0.800 ± 0.010 g was weighed and added into a fecal sample box, processed using a HALO-F100 fecal processor (Suzhou Hailu Medical Technology Co., Ltd., Suzhou, China), and used to prepare a 10% fecal suspension. Thereafter, 2 mL of the supernatant was extracted, after which a short-chain fatty acid pretreatment solution was added, and the solution was frozen at −30 °C for 24 h. After thawing, the solution was centrifuged at 10,000 rpm for 3 min at 4 °C, and the supernatant was removed and filtered through a 0.22-μm filter before analysis.

The amounts of acetic, propionic, isobutyric, butyric, isovaleric, and valeric acids in the culture filtrates were determined using a gas chromatograph (GC-2010 Plus; Shimadzu, Kyoto, Japan) equipped with a DB-FFAP column (Agilent Technologies, Santa Clara, CA, USA) and an H_2_ flame ionization detector. Crotonic acid (*trans*-2-butenoic acid) was used as the internal standard.

### 2.5. 16s rRNA Gene Sequencing

Bacterial 16s rRNA genes were amplified from the extracted DNA using the barcoded primers 338F (5′-ACTCCTACGGGAGGCAGCA-3′) and 806R (5′-GGACTACHVGGGTWTCTAAT-3′). Subsequently, next-generation sequencing was performed using an Illumina HiSeq 2500 system (Illumina, San Diego, CA, USA). Thereafter, sequences were identified by barcodes using the Quantitative Insights into the Microbial Ecology pipeline. A 97% similarity cutoff was used to define operational taxonomic units (OTUs) using mothur. One sequence was selected from each OTU as a representative sequence. The representative sequences were classified using the Ribosomal Database Project classifier method and the SILVA database. Good’s coverage, alpha diversities, including the Simpson and Shannon indices, and richness (observed number of OTUs) were calculated using mothur. All sequences are available in the SRA database with accession number SRP107933.

### 2.6. Metagenomic Measurements

The qualified DNA fragments were disrupted using a Covaris focused sonicator (Covaris Inc., Woburn, MA, USA), followed by end-terminal repair and purification of DNA fragments. Next, sequencing adapters were attached to both ends of DNA fragments and PCR amplification was performed to recover target fragments, after which quality control and quantification of sample libraries were performed. Qualified libraries were sequenced using an Illumina platform.

### 2.7. Metagenome Bioinformatics Analysis

Functional annotation and abundance analyses of metabolic pathways (KEGG), homologous gene clusters (eggNOG), and carbohydrate-active enzymes (CAZy) were performed based on the gene catalog. Based on species abundance and functional abundance tables, abundance clustering analysis, PCA, LEfSe multivariate statistical analysis, and comparative analysis of metabolic pathways were performed to uncover differences in species and functional composition between high- and low-gas-producing groups of bacteria.

### 2.8. Bacterial Isolation and In Vitro Fermentation

Media enriched with gas-producing bacteria with relatively high gas production in fermentation flasks were serially diluted, and then the bacteria were plated on culture plates, which were subsequently placed on an anaerobic bench for 72 h at 37 °C. After incubation, single colonies were randomly transferred to PYG media and incubated anaerobically at 37 °C, whereas MALDI-TOF was performed to identify the bacterial species. Afterward, PYG media was used to prepare bacterial culture broth, and pure cultures of different strains (*E. coli*, *K. pneumoniae*, *C. perfringens*, and *B. longum*) were inoculated into medium containing lactose, FOS, GOS, or 2′-FL, and cultured. After 24 h of fermentation, the air pressure of the culture flasks and the amount of H_2_, CO_2_, H_2_S, and CH_4_ were measured.

### 2.9. Determination of Gas Pressure and Composition

Gas pressure was analyzed using an HT-1890 manometer (Dongguan Xintai Instrument Co. Ltd., Dongguan, China). The gas composition analysis was carried out using equipment jointly developed by Hangzhou Hailu Medical Technology Co., Ltd. (Hangzhou, China) and the present study’s team. The equipment consists of a gas pump and detection module. The detection module measures H_2_ and H_2_S using an electrochemical sensor and CO_2_ and CH_4_ using an infrared sensor.

### 2.10. FITC-2′-FL Preparation

Xi’an Qiyue Biotechnology Co., Ltd. was commissioned to conjugate fluorescein isothiocyanate (FITC) and 2′-FL by chemical modification to form a stable FITC-2′-FL labeled product, and the free FITC was removed using a dialysis bag. The labeled product was lyophilized and analyzed using high-performance liquid chromatography. Finally, the quality-qualified FITC-2′-FL was dissolved in PBS at a concentration of 15 mg/mL and stored at −20 °C. 

### 2.11. Bacteria-Organoid Interaction Experiments

#### 2.11.1. Bacterial Culture and Quantification

Laboratory isolates of *K. pneumoniae* and *C. perfringens* were inoculated in the appropriate liquid media and incubated at 37 °C with 5% CO_2_ (an anaerobic pack was used for anaerobic bacteria). After the liquid medium became turbid, the OD_600_ of the broth was measured using a microplate reader. The number of bacteria in the broth was calculated in CFU/mL based on a previously plotted regression curve.

#### 2.11.2. Conversion of Intestinal Mucosal Organoids into 2D Cells

Matrigel was mixed with 1× D-PBS at a ratio of 1:49 and 100 µL of matrix gel-DPBS mixture was added into Transwell chambers, or 24 mm × 24 mm cell slides were placed into 6-well plates and mixed with 1000 µL of the matrix gel-DPBS mixture for incubation, or 500 µL of the matrix gel-DPBS mixture was added into 12-well plates and incubated for at least 1 h at 37 °C. When the intestinal mucosal organoids grew to a suitable size, the old medium was discarded, and 1 mL of TrypLE™ Express enzyme (Thermo Fisher Scientific, Waltham, MA, USA) containing 0.5 mM EDTA was added to each well to harvest the organoids. After pipetting up and down a few times, the organoids were incubated in a 37 °C water bath for 5 min. If cell clumps were still visible, a 1000-µL pipette was used for pipetting up and down to separate the clumps. The digestion reaction was terminated using 1 mL DMEM containing 10% FBS, the cells were then passed through a 100-µm cell filter, and 1× D-PBS was added to 10 mL. Thereafter, the suspension was centrifuged at 300× *g* for 5 min. The supernatant was discarded, and the cell pellet was resuspended in growth medium (AB solution) containing 10 µM Y-27632. The Transwell was supplemented with 100 µL of growth medium (AB solution) containing 10 µM of Y-27632 growth medium, and the lower chamber was supplemented with 600 µL of growth medium. The differentiation medium was changed after 2 days of incubation at 37 °C in a 5% CO_2_ incubator. 

The study and operation procedures were in accordance with the ethical standards specified by the Clinical Research Committee of the First Affiliated Hospital, College of Medicine, Zhejiang University (The certificate No. 2020 IIT, Consent letter No. 562).

#### 2.11.3. Bacterial Adhesion Experiments

Cells were inoculated into 12-well plates at a density of 1 × 10^5^, with 1 mL of medium containing 2 mg/mL 2′-FL in half of the wells and 1 mL of medium without 2′-FL in the other half. The cell culture plate was placed in a 37 °C and 5% CO_2_ incubator for 48 h. The cell surface was then washed 2–3 times with PBS, and 0.5 mL of fresh medium without penicillin/streptomycin and 2′-FL was added. Afterward, *K. pneumoniae* and *C. perfringens* were added to a multiplicity of infection (MOI) of 20, i.e., 20 bacteria per cell. The cell culture plates were subsequently placed in an incubator at 37 °C in 5% CO_2_ for 1 h. Thereafter, the medium was discarded, and the cell surface was washed thrice with PBS, after which sterile water (1 mL) containing 1% Triton X-100 was added to each well and incubated at room temperature for 20–30 min. The cell lysates were then mixed by pipetting and diluted 10^−2^- and 10^−3^-fold. Next, three corresponding agar plates were spread at each concentration with a suspension volume of 100 µL. Finally, the culture plates were incubated overnight at 37 °C in 5% CO_2_ and enumerated to obtain CFU/mL.

#### 2.11.4. Bacterial DNA Extraction

An overnight bacterial suspension (0.5 mL) was added to a 1.5-mL centrifuge tube and centrifuged at 8000 rpm for 1 min at room temperature. A buffer for digestion (180 µL) was added, followed by 20 µL of proteinase K, and the solution was mixed evenly. The solution was then incubated in a 56 °C water bath for 1 h until complete lysis was achieved. Next, 200 µL of Buffer BD was added, and the tube contents were mixed well by inversion before incubation in a water bath for 10 min at 70 °C. Subsequently, absolute ethanol (200 µL) was added, and the tube contents were mixed thoroughly by inversion. A column was then placed in a collection tube, and the solution and translucent fibrous suspension were pipetted into the column, which was left to stand for 2 min before centrifugation at room temperature and 12,000 rpm for 1 min. Afterward, the waste solution in the collection tube was discarded, and the column was placed back into the collection tube. Next, PW solution (500 µL) was added, and the tube was centrifuged at 10,000 rpm for 30 s before the filtrate was discarded. Thereafter, the column was placed back into the collection tube and centrifuged at 12,000 rpm for 2 min at room temperature to remove the residual wash solution. The column was transferred into a new 1.5-mL centrifuge tube, then 50–100 µL CE Buffer was added, and the tube was left to stand for 3 min before centrifugation at 12,000 rpm for 2 min at room temperature to collect the DNA solution. Finally, the extracted DNA was used immediately for the next experiment or stored at −20 °C.

#### 2.11.5. Cell Slide

A 24 mm × 24 mm coverslip was added to a 6-cm cell culture dish. Next, 6 mL of PBS containing 2% penicillin/streptomycin was added, and the culture dish was placed on a clean bench and incubated at room temperature for more than 2 h. The coverslip surface was then washed with PBS thrice, and forceps were used to place the coverslip in a 6-well plate before washing once with PBS. Thereafter, 1 mL of PBS containing 1% penicillin/streptomycin was added, after which 2D organoid cells at 90% confluence were harvested, and the cells digested into single cells using trypsin. After centrifugation, the cells were resuspended in a single-cell suspension using the appropriate medium. The PBS in the 6-well plate was discarded, and 3 × 10^5^ cells were added to each well. Afterward, 2′-FL was added at a final concentration of 2 mg/mL in the treatment wells, after which the corresponding medium was added to a total of 2 mL of medium in a 6-well plate. The cell culture plate was finally incubated at 37 °C in a 5% CO_2_ incubator.

#### 2.11.6. Fluorescence In Situ Hybrid (FISH) Staining of Bacteria

The target cell slides were added into 6-well plates containing 24 mm × 24 mm coverslips. Half of the wells were filled with 2 mL of medium containing 2 mg/mL 2′-FL and the other half with 2 mL of medium without 2′-FL. The cell culture plate was then placed in a 37 °C and 5% CO_2_ incubator for 48 h. Afterward, the cell surface was washed with PBS 2–3 times, and then fresh medium (2 mL) without penicillin/streptomycin and 2′-FL was added to *K. pneumoniae* and *C. perfringens* to an MOI of 20. Next, the cell culture plates were incubated at 37 °C in a 5% CO_2_ incubator for 1 h, after which the medium was discarded, and the cell surface was washed thrice with PBS. Finally, an in situ hybridization fixative (2 mL) was added, and the plate was incubated for 30 min at room temperature away from light. 

#### 2.11.7. FITC-2′-FL Localization

The target cell slides were added into a 6-well plate containing 24 mm × 24 mm coverslips, after which 2 mL of the corresponding medium was added to the cells in each well. The cell culture plate was then placed in a 37 °C and 5% CO_2_ incubator for 48 h, and the cell surface was subsequently washed with PBS 2–3 times. Thereafter, 2 mL of fresh medium without penicillin/streptomycin was added, and half of the wells were treated with 0.1 mg/mL of FITC-2′-FL, whereas the other half was left untreated. In all wells, *K. pneumoniae* and *C. perfringens* was added to an MOI of 20. Subsequently, the cell culture plates were incubated at 37 °C in a 5% CO_2_ incubator away from light for 1 h. The medium was then discarded, and the cell surface was washed thrice with PBS. Afterward, an in situ hybridization fixative (2 mL) was added, and the plates were incubated for 30 min at room temperature away from light. Finally, using glycerol (50%) for mounting, fluorescence staining was observed using a fluorescence microscope.

### 2.12. Transcriptome Sequencing and Bioinformatics Analysis

After extracting the total RNA from the bacterially infected organoids and digesting the DNA with DNase, the eukaryotic mRNA was enriched using oligo (dT)-containing magnetic beads, and the mRNA was broken into short fragments by adding a disruption reagent. The disrupted mRNA was used as a template to synthesize single-stranded cDNA using random hexamer primers, and double-stranded cDNA was synthesized by preparing a double-stranded reaction system. Double-stranded cDNA was purified using an assay kit followed by end-terminal repair and polyA tail and sequencing adapter ligation. This was followed by fragment size selection and PCR amplification, and the quality of the constructed library was examined using an Agilent 2100 Bioanalyzer (Agilent Technologies) and sequenced using the Illumina HiSeq™ 2500 sequencer (Illumina), generating 125 or 150 bp double-end data.

Clean reads were aligned to the reference genome using hisat2 to obtain information on the position on the reference genome or gene, as well as information on sequence characteristics specific to the sequenced samples. The number of counts for each sample gene was normalized using DESeq software (base mean values were used to estimate expression), the fold difference calculated, and the number of reads was tested for significant differences using the negative binomial distribution test. Finally, the fold difference tests and tests for differences between treatments (log_2_ (fold-change) > 1 and *p* < 0.05, respectively) were used to screen for differential protein-coding genes. GO enrichment and KEGG pathway enrichment analyses were performed to determine the biological functions or pathways primarily affected by the differential genes.

### 2.13. Statistical Analysis

All data were analyzed using SPSS v. 23.0 (SPSS Inc., Chicago, IL, USA) and expressed as the mean ± standard deviation. Comparisons between multiple groups were performed using one-way ANOVA and tested for homogeneity of variance. Correlation analysis was performed using Pearson’s correlation coefficient. Images mapping were performed with R language and the software of GraphPad Prism. Statistical significance was set at *p* < 0.05, and *p* < 0.01 indicated highly statistically significant differences.

## 3. Results

### 3.1. Analysis of Differences in the Intestinal Flora Structure of Infants and Toddlers Fed Differently and at Different Ages

Differences in fecal bacterial genus levels between breastfed (BF) and formula-fed (MF) infants in different age groups (growth stages 1, 2, and 3) were compared. The results showed that the species *E. coli*, and the genera *Klebsiella*, *Bacteroides*, *Veillonella*, and *Bifidobacterium* were dominant in the feces of infants, regardless of whether they were BF or MF (Figure 1). However, there were significant differences between genera according to age and feeding method. In growth stage 1, *Klebsiella* was the dominant genus, and *E. coli* dominance gradually increased until it became the dominant species. However, at growth stages 2 and 3, *Klebsiella* levels were significantly higher in the fecal bacteria of MF infants than in the BF group compared with those in growth stage 1. In growth stages 1 and 2, *Bifidobacterium* levels were significantly higher in BF infants than those in MF infants, but by growth stage 3, they were higher in the MF group than in the BF group.

### 3.2. Comparison of In Vitro Simulated Fermentation Gas Production by Fecal Bacteria in Infants and Toddlers Fed Differently and at Different Growth Stages

The transfer of infant feces to media containing different carbon sources (lactose, fructooligosaccharide (FOS), galactooligosaccharide (GOS), and fucosyllactose (2′-FL)) showed that gas production by infant fecal bacteria from MF sources was significantly higher than that of the control group (YCFA). In contrast, among BF infant fecal bacteria, only the in vitro fermentation of fecal bacteria at growth stages 1 and 2 produced more gas than the control, whereas gas production at growth stage 3 did not differ significantly between media. In addition, the gas production of infant fecal bacteria from MF infants was significantly (*p* < 0.05) higher than that produced by BF infant fecal bacteria in media with different carbon sources (Figure 2).

### 3.3. Comparison of In Vitro Simulated Fermentation of Fecal Ammonia Levels in Infants and Toddlers Fed Differently and at Different Growth Stages

In vitro incubation of fecal bacteria from infants and toddlers at different growth stages in media containing different carbon sources showed that mean fecal ammonia levels were higher in the MF group than in the BF group, regardless of carbon source in the medium. The control medium without a carbon source (YCFA) had the highest fecal ammonia levels, followed by that with 2′-FL, whereas the culture media with the other three carbon sources had lower fecal ammonia levels. This shows that the YCFA medium had to supply energy through amino acid catabolism due to the absence of a carbon source, resulting in the highest level of fecal ammonia among the treatments. The level of fecal ammonia in the 2′-FL group was higher than that for the other three carbon sources (lactose, FOS, and GOS), which may be due to the lower utilization of 2′-FL by the fecal bacteria. At each corresponding time point, fecal ammonia was, on average, higher in the MF group than in the BF group, which may be related to the differences in the structure of the two groups (Appendix A).

Further cluster heat map analysis was conducted on all fecal samples based on gas and ammonia production. The results showed that gas production was highly correlated with ammonia production, and fecal samples with high gas production also had high ammonia production (Figure 3).

### 3.4. Comparison of In Vitro Simulated Fermentation of Short-Chain Fatty Acid (SCFA) Production by Fecal Bacteria in Infants and Toddlers Fed Differently and at Different Growth Stages

When infant fecal bacteria were transferred to in vitro culture media containing different carbon sources, in vitro cultures of MF fecal bacteria produced higher levels of acetic, propionic, and butyric acids than BF fecal bacteria, regardless of carbon source. When lactose, FOS, and GOS were used as carbon sources in the medium, the flora primarily produced acetic, propionic, butyric, and valeric acids, with little isobutyric and isovaleric acid production (Appendix A). In contrast to the medium with supplementary carbon sources, the control YCFA medium produced more isobutyric and isovaleric acids, which were primarily produced through amino acid catabolism rather than carbon metabolism. When the medium used 2′-FL as a carbon source, microflora fermentation produced not only acetic, propionic, butyric, and valeric acids but also isobutyric and isovaleric acids, which were combined with the data on the in vitro culture of fecal ammonia metabolic characteristics (Figure 3), further indicating that infant fecal bacteria do not utilize 2′-FL as efficiently as other carbon sources. This implies that the colon obtains energy from amino acid catabolism, in turn metabolizing significantly higher levels of isobutyric and isovaleric acids.

### 3.5. Analysis of the Correlation between Differences in Gas Production by Feeding Method and Differences in Bacterial Genus

The original fecal bacteria of infants were divided into high- and low-gas-producing groups according to the in vitro culture gas production of fecal bacteria in infants fed differently. 16S rRNA sequencing results were combined with those of differences in genera between the two groups (heat map, boxplot, LEfSe, and PC analyses). These results showed that the high-gas-producing group of infant fecal bacteria was enriched with Enterobacteriaceae and *Klebsiella*, whereas the low-gas-producing group was enriched with *Bifidobacterium* and *Bacteroidetes* (Figure 4).

Correlation analysis based on the gas production during in vitro fermentation of fecal bacteria from infants and toddlers fed differently and at different growth stages, as well as flora abundance, showed that gas production during in vitro fermentation of fecal bacteria negatively correlated with *Bifidobacterium* regardless of the growth stage (Appendix A). Furthermore, gas production in media with different carbon sources negatively correlated with *Bifidobacterium* regardless of the feeding pattern and positively correlated with *Klebsiella* (Appendix A). However, the correlation was not significant for media containing 2′-FL as the carbon source (Appendix A). 

### 3.6. Analysis of Structural and Functional Genetic Differences in Bacterial Populations in Different Gas-Producing Media

Rather different sources of bacteria were used to ferment different carbon sources using an in vitro fermentation simulation system with FOS and 2′-FL as carbon sources, respectively, and the fermentation microorganisms were subjected to metagenome sequencing. Barplot analysis showed that the MF infant fecal bacteria had a higher *Klebsiella* and *Perfringens* abundance than that of BF infant fecal bacteria. The strain with the highest abundance in the BF group was *Bifidobacterium* (Figure 5).

The results of the metagenomic eggNOG analysis showed that flora from BF infants were enriched in more functional genes than those from MF infants (Appendix A). According to the CAZy database, carbohydrate metabolism-related genes were enriched in the metagenomic data with FOS and 2′-FL as carbon sources. From the heat map (Appendix A), it can be seen that there were significant differences in the enrichment of genes related to oligosaccharide metabolism in infant fecal bacteria cultured in medium with two different carbon sources, with the 2′-FL group enriched in multiple enzyme families for human milk oligosaccharide metabolism, including the carbohydrate-binding module (CBM) and plasmin (PL) families (Appendix A).

### 3.7. Analysis of Structural and Functional Genetic Differences in the Microflora in Different Gas-Producing Media

*E. coli*, *K. pneumoniae*, *C. perfringens*, and *B. longum*, were isolated and cultured separately from infant fecal bacteria in culture media containing different carbon sources. The results showed that none of the bacteria could efficiently use 2′-FL for proliferation compared with the control YCFA medium (Figure 6A); *K. pneumoniae* and *C. perfringens* had the highest gas production, followed by *E. coli*. When the medium was supplied with 2′-FL as the sole carbon source, gas production was significantly (*p* < 0.05) lower than that of the medium containing other carbon sources (Figure 6B). *Bifidobacterium* did not produce gas regardless of the carbon source, and *K. pneumoniae* and *C. perfringens* utilized GOS, FOS, and lactose better than *E. coli* (Figure 6B).

Further analysis of the production of each type of gas in the aforementioned monobacterial fermentation media showed that only *E. coli*, *K. pneumonia*, and *C. perfringens*, not *Bifidobacterium*, produced CO_2_, H_2_, and H_2_S (Figure 7).

### 3.8. Effect of 2′-FL on Adherence and Expression of Genes Associated with K. pneumoniae and C. perfringens Adhesion to Organoids

In vitro cultures of *K. pneumoniae* and *C. perfringens*, infecting human colonic epithelial organoids, were used, and the results showed that adding 2′-FL to the medium significantly (*p* < 0.05) reduced the adhesion of *K. pneumonia* (*p* < 0.0001) and *C. perfringens* (*p* < 0.05) to the intestinal epithelial layer, respectively (Figure 8).

Transcriptome analysis showed that the genes associated with the cellular binding protein factor pathway (SERF1A and MARK2P17) (Appendix A) were significantly up-regulated when the organoids were infected by *K. pneumoniae*, and were significantly down-regulated when they were co-cultured with 2′-FL (Figure 9A). The expression of pathway genes (TIRB3, ADM2, PCK2) (Appendix A) that bind to ubiquitin-like protein ligase and transcriptional corepressor activity were significantly down-regulated after infection by *C. perfringens*, and significantly up-regulated after 2′-FL treatment (Figure 9B). It is suggested that there may be different mechanisms for different strains to infect organoids and prevent 2′-FL from adhering to organoids.

## 4. Discussion

Numerous preliminary studies have demonstrated that the human intestine is an open fermentation system. Analysis of parameters such as pH, redox potential, the ratio of carbon to nitrogen sources, and emptying rate in various anatomical parts of the human GI tract verified that most of the intestine is characterized by continuous anaerobic fermentation [29]. Controlling the fermentation parameters and media composition in an in vitro gut microbial simulation system enables the maintenance of normal intestinal bacteria growth in vitro. In our laboratory, we have established an in vitro simulated intestinal fermentation system and demonstrated through extensive experiments that 60–80% of bacterial species in the guts of Chinese individuals can grow in a simulated in vitro intestinal fermentation system [29,30]. By adjusting the culture medium and fermentation conditions, *Bacteroides* and *Prevotella* have been successfully cultured in an in vitro intestinal simulation system. Furthermore, by connecting a gas analyzer on top of the in vitro simulation model, the production and composition of intestinal bacterial gases can be studied under different fermentation conditions, especially under pre- and probiotic supplementation conditions. This makes it possible to design experiments for exploring the relationship between intestinal flora gas metabolite-digestion-related diseases [31].

Various digestive tract diseases and diseases associated with metabolic abnormalities have been associated with abnormal gas production due to dysbiosis [32]. For example, an increase in the number of sulfate-reducing bacteria in the intestine causes the production of high H_2_S concentrations, which subsequently cause DNA damage in the host intestinal epithelium and disrupts the intestinal mucosal barrier of the host [24]. Several studies have demonstrated a significant increase in H_2_S in the stools of patients with ulcerative colitis, indicating that high H_2_S concentrations may play an important role in causing mucosal inflammation and carcinogenesis [24,33,34,35]. Furthermore, studies overseas have revealed that increased methane content in the intestine and methanogens affect intestinal motility and are associated with constipation [36,37]. There are also sporadic reports of increased gas production in the intestines of patients with irritable bowel syndrome (IBS) [38].

Neonatal abdominal distension or infantile abdominal distension is a common and more difficult symptom to manage clinically [20]. Neonatal abdominal distension is predominantly gas distension and can be caused by various reasons, including improper feeding, indigestion, abdominal infection, perforation, and neonatal necrotizing enterocolitis. However, little is known regarding the effect of feeding methods on gas production by flora. Accordingly, we conducted a systematic study of the effects of breastfeeding and mixed feeding regimes on the structure, metabolism, and gas production of infant intestinal flora. By comparing the differences in fecal bacterial genus levels between BF and MF infants at different ages (≤1 month, 1–3 months, and 3–6 months), we found that there were significant differences in the levels of intestinal flora genera among infants, with BF infants having significantly higher levels of *Bifidobacterium* than those of MF infants in the early stages of life (stages 1 and 2). However, the MF group had higher levels of *Bifidobacterium* than those of the BF group in growth stage 3. This may be related to the fact that the factors that promote the proliferation of *Bifidobacteria* in the breast milk of exclusively breastfed infants were insufficient or decreased during the latter stages of lactation compared with MF. Breast milk contains approximately 5–15 g/L of HMOs, the third most abundant solid component after fat and lactose. HMOs have multiple health benefits such as the selective proliferation of probiotics, improvement of intestinal flora and intestinal barrier function, maturation of the immune system, and enhancement of brain and cognitive development in infants [39]. 2′-FL is one of the major HMOs found in breast milk and can stimulate the proliferation of *Bifidobacterium* and *Bacteroides* in the infant intestine, although its levels decrease with advancing lactation [40]. This suggests that appropriate supplementation with milk powder during the late lactation period can be helpful for the proliferation of *Bifidobacteria*.

Culturing infant feces in media containing different carbon sources (lactose, FOS, GOS, and 2′-FL) showed that infant fecal bacteria from MF infants resulted in significantly higher gas production than that of the control group (YCFA). Gas production of infant intestinal flora negatively correlated with both *Bifidobacterium* and *Klebsiella*; however, the correlation was not significant for medium containing 2′-FL as the carbon source. It is possible that formula feeding contributes to increased gas production in the infant intestine compared with breastfeeding. Further isolation of individual bacteria and their in vitro fermentation showed that neither *E. coli*, *K. pneumoniae*, nor *B. longum* could efficiently utilize 2′-FL for proliferation. *Bifidobacterium* did not produce gas regardless of the carbon source in the culture medium. Except for *Bifidobacterium*, all three bacteria produced CO_2_, H_2_, and H_2_S, consistent with previous reports [24]. The major gas components in the human gut are N_2_ (59%), H_2_ (20.9%), CO_2_ (9%), CH_4_ (7.2%), O_2_ (3.9%), and H_2_S (0.00028%) [19]. Most gases in the intestine, including H_2_, CO_2_, H_2_S, and CH_4_, are bacterial metabolites, except for a small proportion of CO_2_ and H_2_S produced by the body itself [19]. Adults can produce approximately 12 L of H_2_ per day, and 99% of this is produced by bacterial fermentation in the colon [32,41]. If this large amount of gas is not consumed through normal metabolic pathways, it will undoubtedly cause flatulence and discomfort in the human intestine [42].

Some *Bifidobacterium* species can utilize 2′-FL with varying efficiency [43]; the ability of *Bifidobacteria* to utilize 2′-FL is related to whether the bacteria possess certain key genes in their genomes [44]. In the present study, using the CAZy database, we found that infant fecal bacteria that could utilize 2′-FL were enriched with several enzyme families specific to human milk oligosaccharide metabolism, including the CBM and PL families.

Fecal ammonia, SCFA, and gas production results after in vitro fermentation showed that infant intestinal flora might not utilize 2′-FL as efficiently as other carbon sources such as lactose, FOS, and GOS (Figure 6). This shows that 2′-FL may have other important biological functions as it is considered a very important HMO. The structure of 2′-FL is similar to that of glycoproteins on intestinal epithelial cells, part of the sugar chain, and can act as decoy receptors to bind pathogenic bacteria in the intestine, in turn preventing pathogenic bacteria from binding to the intestinal epithelial cell receptors and successfully colonizing the intestine. Ruiz-Palacios et al. showed that the structure of 2′-FL effectively inhibited *Campylobacter jejuni* colonization in mice, which causes diarrhea in infants and whose binding to intestinal H-2 antigen is a prerequisite for host cell infection [45]. Furthermore, Stefan et al. used bioengineered 2′-FL to effectively inhibit *C. jejuni* and enteropathogenic *E. coli* from adhering to the Caco-2 human enterocyte line in an in vitro assay, achieving 26 and 18% inhibition, respectively [46]. Ingestion of 2′-FL reduced colonization by *C. jejuni* by 80% and attenuated the histological features of intestinal inflammation in a mouse model inoculated with *C. jejuni* [47]. Observational experiments have also demonstrated that high 2′-FL levels in breast milk attenuate infantile diarrhea caused by *C. jejuni* in a dose-dependent manner [48].

In the present study, we used in vitro cultured *K. pneumoniae* and *C. perfringens* to infect human colonic epithelial organoids, and the results showed that adding 2′-FL to the culture medium significantly reduced the adhesion of *K. pneumoniae* and *C. perfringens* to the intestinal epithelial layer (Figure 8). Moreover, transcriptome analysis showed that there may be different mechanisms for different strains to infect organoids and prevent 2′-FL from adhering to organoids. In an enterotoxin-producing *E. coli* infection of intestinal epithelial cells, Ying et al. showed that 2′-FL reduced lipopolysaccharide-induced inflammation by downregulating the expression of lipopolysaccharide receptor (CD14) on the surface of intestinal epithelial cells and inhibiting the release of pro-inflammatory signaling molecules [49]. In addition, Holscher et al. evaluated the effects of 2′-FL on enterocytes in vitro and showed that 2′-FL reduced enterocyte proliferation and promoted the differentiation ability of enterocytes [50]. Thus, 2′-FL exerts immunomodulatory effects in the intestine. As well as its immunomodulatory effects in the intestine, some 2′-FL can be transported across the intestinal epithelium via receptor-mediated endocytosis and the paracellular pathway into the circulatory system, where it acts as an immune factor throughout the body. In a mouse model of ovalbumin allergy, 2′-FL administration by gavage reduced food allergy symptoms by inducing T regulatory cells and indirectly stabilizing mast cells [51]. Clinical trials have also shown that formula supplemented with 2′-FL resulted in plasma levels of inflammatory cytokines that were more similar to those of exclusively BF infants [52]. It may be related to the role of 2′-FL in strengthening and supporting immune development and regulation.

In summary, 2′-FL may reduce gas production in the infant intestine through two mechanisms: first, it cannot be used by harmful intestinal bacteria to produce gas, and second, it can inhibit colonization of the intestinal mucosa by harmful bacteria by regulating the expression of intestinal epithelial pathogenic genes/signaling pathways, thus reducing the proliferation of gas-producing harmful bacteria in the intestine. This study systematically revealed the possible mechanism of different feeding patterns on infant intestinal gas production, which has positive guiding significance for the future research and development of infant formula milk powder.

## Figures and Tables

**Figure 1 nutrients-14-01568-f001:**
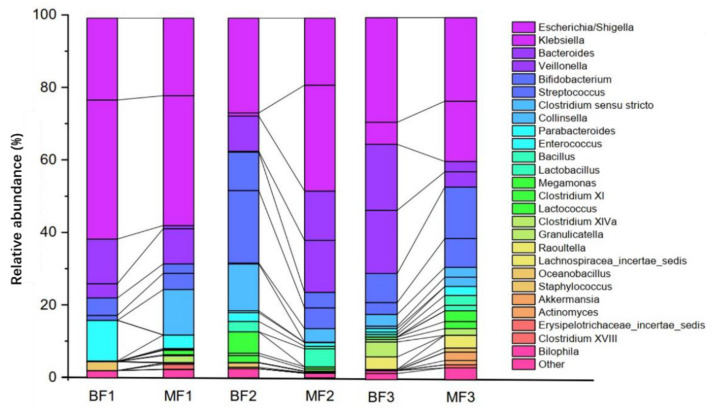
Comparison of fecal bacteria structure among infants and toddlers fed differently and at different ages (BF, breastfeeding; MF, mixed feeding).

**Figure 2 nutrients-14-01568-f002:**
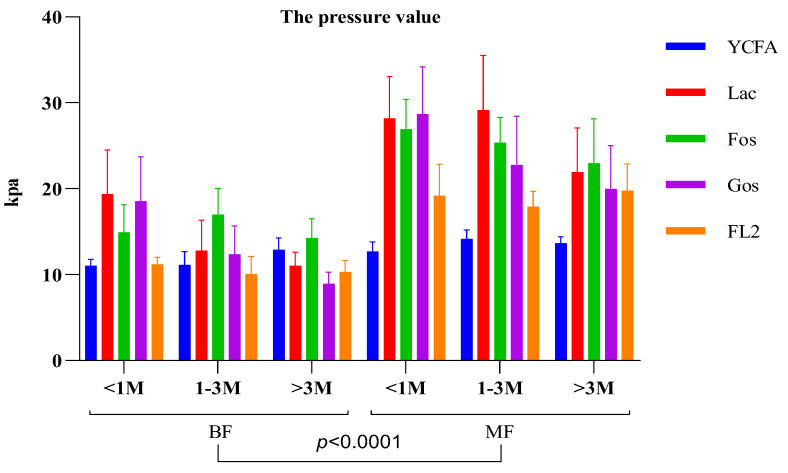
Comparison of in vitro simulated fermentation gas production by fecal bacteria in infants and toddlers fed differently and at different growth stages. Data were expressed as the mean ± SEM.

**Figure 3 nutrients-14-01568-f003:**
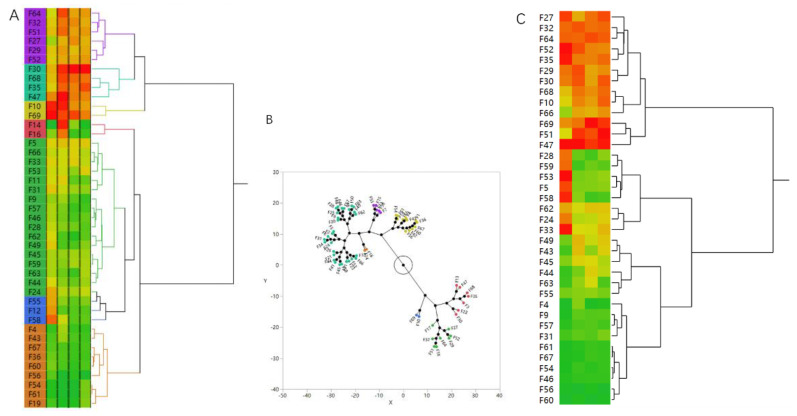
Cluster heat map analysis of fecal samples based on gas and ammonia production (**A**): the cluster (**A**) and constellation (**B**) diagram of gas production; the cluster diagram of ammonia production (**C**).

**Figure 4 nutrients-14-01568-f004:**
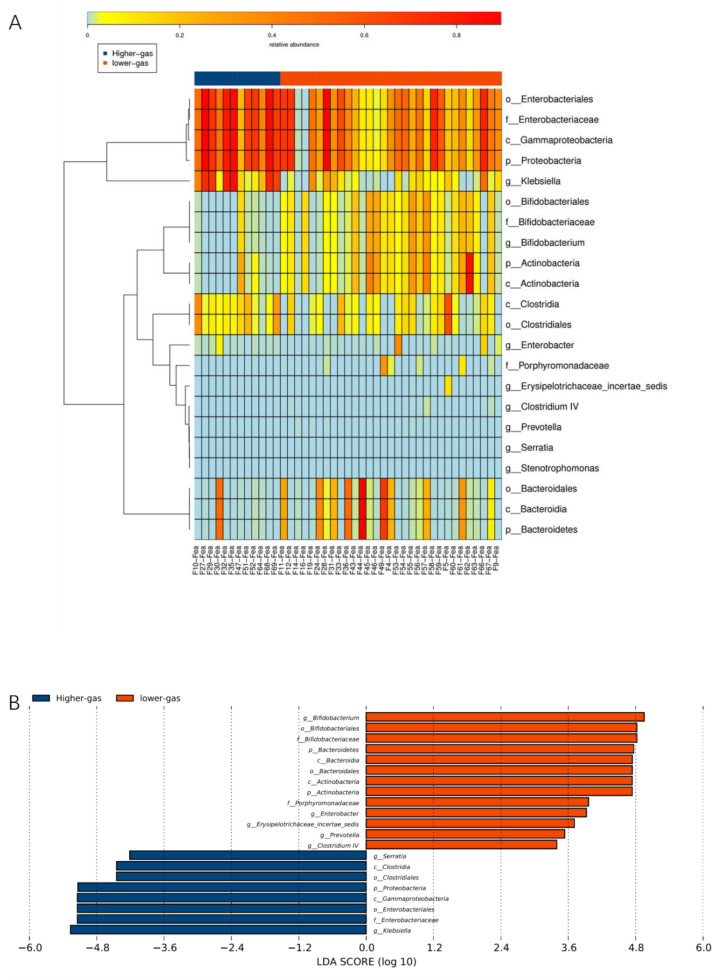
Analysis of differences in gas production between feeding methods and genus differences ((**A**) heat map, (**B**) Boxplot, (**C**) PCA, (**D**) PC analyses).

**Figure 5 nutrients-14-01568-f005:**
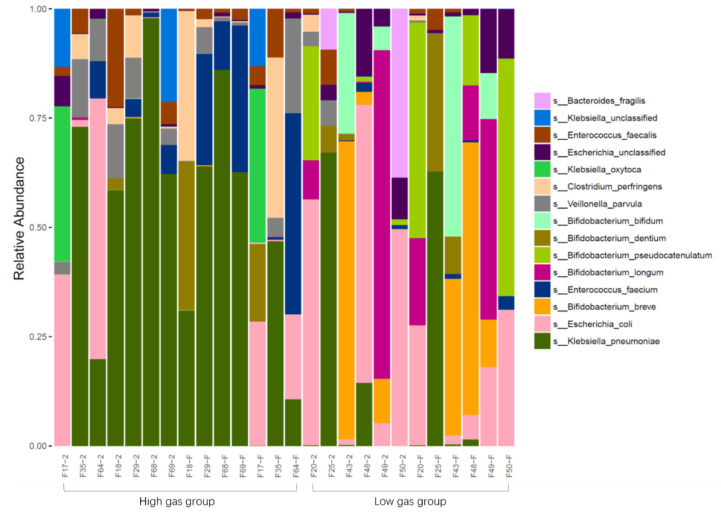
Analysis of structural metagenomic differences in fecal bacteria in different gas production media. Each infant fecal bacteria were cultured in medium with two different carbon sources (2′-FL and FOS).

**Figure 6 nutrients-14-01568-f006:**
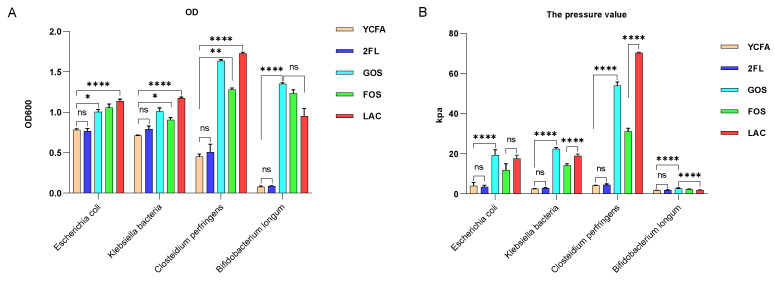
Effect of different carbon sources on the proliferation (**A**) and gas production of different bacteria (**B**). Both bacterial proliferation (OD value) and the pressure value were expressed as the mean ± SEM. Differences are considered significant at *p* < 0.05 (*), *p* < 0.01 (**) and *p* < 0.0001 (****).

**Figure 7 nutrients-14-01568-f007:**
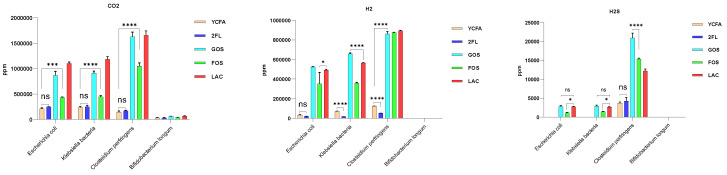
Analysis of the types of in vitro fermentation gases produced by different bacterial strains. The production of CO_2_, H_2_, and H_2_S were expressed as the mean ± SEM. Differences are considered significant at *p* < 0.05 (*), *p* < 0.001 (***), and *p* < 0.0001 (****).

**Figure 8 nutrients-14-01568-f008:**
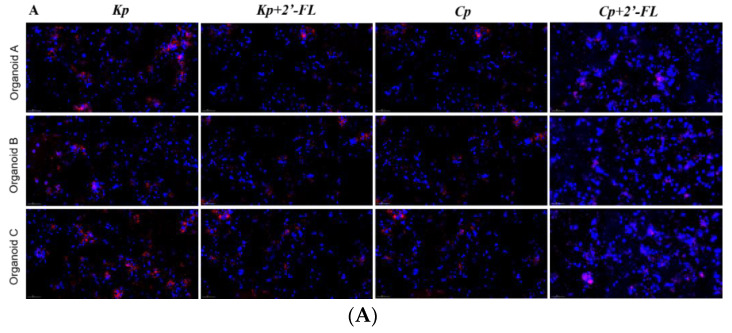
Analysis of the effects of 2′-FL on *Klebsiella pneumoniae* and *Clostridium perfringens* adhesion to organoids in fluorescence detection images (**A**) and qFISH (**B**). The scale bar is 50 μm. Data were expressed as the mean ± SEM.

**Figure 9 nutrients-14-01568-f009:**
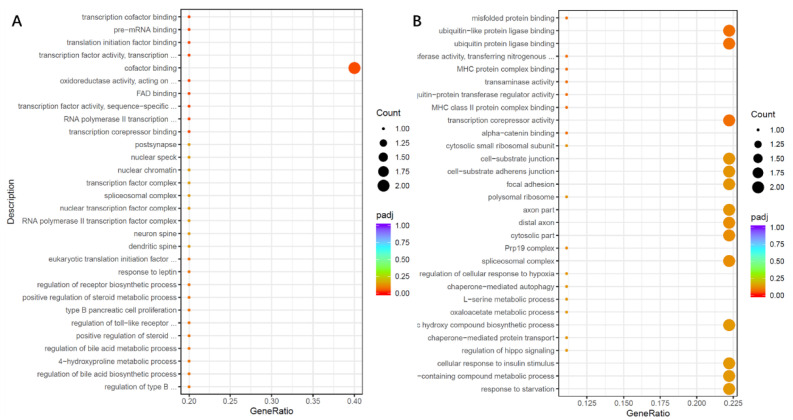
Effect of 2′-FL on the signaling pathways of *Klebsiella pneumoniae* (**A**) and *Clostridium perfringens* (**B**) after organoid infection.

## Data Availability

The authors declare that the data supporting the findings of this study are presented in the manuscript. The sequences obtained in this study of 16S, and metagenomic and transcriptome data submission numbers were deposited in the NCBI Sequence Read Archive under the accession number PRJNA 778443, PRJNA778457, and PRJNA771527, respectively. Additional data sources are also available from the corresponding author upon request.

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
