# Peer review of "Effects of Different Feeding Methods on the Structure, Metabolism, and Gas Production of Infant and Toddler Intestinal Flora and Their Mechanisms"

_nutrients, 2022, doi:10.3390/nu14081568_

Round 1
Reviewer 1 Report
I find the idea for the research project described in this manuscript quite interesting. But the presentation of results, and in consequence some of the conclusions, should be grately improved.
Line 41-50 Rephrase in order to make consistent. Firmicutes and Bacteroidetes Or Bifidobacteria dominating the gut microflora at the end of the first month ?
Lines 202, 222,253, 465 , 468 , 472, 606 (not all instances were denoted here), C. perfringens
Figure 3. Please add description of columns in the heatmaps. Moreover, the colour coding in the left-hand heatmap should be explained. Should not the other heatmap receive colouring. Using the same colours as in the other graph could help to visualize overlap between clusters, The caption to Fig. 3 should include description of the updated graphs.
Figure 4. Caption - Graphs (A,B,C, D) of the figure are not properly described, eg. Fig4C shows PCA plot and not LEfSe.
Line 444 “Two different sources of infant fecal bacteria (...) were fermented” Rather different sources of bacteria were used to ferment different carbon sources. Correct the sentences.
Line 461 should be “carbohydrate-binding module”
Figure S5 The graph B is very unfriendly to color-blind readers. Caption to this figure is insufficient. There is no information of the graphs A and B
Figure 5. Why the graph shows only 13 pairs of FOS vs 2’-FL metagenomic profiles? Is there a way to know which isolates belong to high- and low-gas producers? Update the figure and its caption.
Line 468 Where is or what is Fig 11?
Line 469 change “cultured” to „supplied”
Figures 6 and 7 use abbreviation “LAT”, which probably stands for lactose. In other Figures this carbon source is marked as “Lac”. At least provide description of abbreviations used in captions to these figures. Which statistics was used to calculate p-values in these plots?
Figure 8 It presents microphotographs of different bacterial strains co-cultured with organoids, but rows are not described. Are these examples of photographs taken? How many were there to calculate the bars in the lower graph. The graph B should use same labels as the microphotographs above. This would allow to present data without the “IG” – which as abbreviation is not determined in the manuscript.
Fragment between lines 492 and 512 (with Fig. 9) – The short description of the results sound interesting, but there are no results shown. The figure 9, which should contain results from transcriptional analysis co-cultured organoids, does not supports the claims made in the text fragment, e.g. downregulation of TIRB3, ADM2, and PCK2 is nowhere to be seen. Moreover, the caption to figure 9 suggest that the illustration presents expression of Klebsiella and Clostridium and needs to be corrected. The figure itself is provided in low resolution and thus it makes reading out of it problematic. This whole fragment needs to be improved. I am not convinced by the presentation of results.
Line 537 “overseas” – where?
Generally, all figure cations need to be improved e.g. by introducing explanation of used abbreviations, labeling of data, etc.
Author Response
Point 1: I find the idea for the research project described in this manuscript quite interesting. But the presentation of results, and in consequence some of the conclusions, should be grately improved.
Response 1: Thank you for your comments. We have revised all suggestions one by one.
Point 2: Line 41-50 Rephrase in order to make consistent. Firmicutes and Bacteroidetes Or Bifidobacteria dominating the gut microflora at the end of the first month ?
Response 2: According to previous reports, bifidobacteria are the dominant bacteria in most neonatal guts at the end of the first month.
Point 3: Lines 202, 222, 253, 465, 468, 472, 606 (not all instances were denoted here), C. perfringens.
Response 3: Thank you for your careful correction. All modifications have been made.
Point 4: Figure 3. Please add description of columns in the heatmaps. Moreover, the colour coding in the left-hand heatmap should be explained. Should not the other heatmap receive colouring. Using the same colours as in the other graph could help to visualize overlap between clusters, The caption to Fig. 3 should include description of the updated graphs.
Response 4: Thank you for your suggestions. The colour coding in the left-hand heatmap was corresponding to the constellation diagram inserted. The caption of figure 3 has been restated.
Point 5: Figure 4. Caption - Graphs (A,B,C, D) of the figure are not properly described, eg. Fig4C shows PCA plot and not LEfSe.
Response 5: Thank you for your suggestions. I have modified the incorrectly description.
Point 6: Line 444 “Two different sources of infant fecal bacteria (...) were fermented” Rather different sources of bacteria were used to ferment different carbon sources. Correct the sentences.
Response 6: Thank you for your suggestion and the modification has been made according to your suggestion.
Point 7: Line 461 should be “carbohydrate-binding module”
Response 7: Thank you for your suggestion and the modification has been made according to your suggestion.
Point 8: Figure S5 The graph B is very unfriendly to color-blind readers. Caption to this figure is insufficient. There is no information of the graphs A and B
Response 8: Thank you for your suggestion. I have clarified the graphics according to your suggestion and divided the original figure into two figures to describe them respectively.
Point 9: Figure 5. Why the graph shows only 13 pairs of FOS vs 2’-FL metagenomic profiles? Is there a way to know which isolates belong to high- and low-gas producers? Update the figure and its caption.
Response 9: Due to the limitation of research funds, this study only tested 14 samples with the highest gas production and 12 samples with the lowest gas production for comparison. In order to make readers more clearly identify, we have remarked the high- and low gas production sample groups in figure 5, respectively.
Point 10: Line 468 Where is or what is Fig 11?
Response 10: Thank you for your correction. I have written Figure 6A instead of Fig 11.
Point 11: Line 469 change “cultured” to „supplied”
Response 11: Thank you for your correction. It has been revised as you suggested.
Point 12: Figures 6 and 7 use abbreviation “LAT”, which probably stands for lactose. In other Figures this carbon source is marked as “Lac”. At least provide description of abbreviations used in captions to these figures. Which statistics was used to calculate p-values in these plots?
Response 12: Thank you for your correction, we have changed the abbreviation of all the lactose letters in the picture to LAC, and the specific statistical content was written.
Point 13: Figure 8 It presents microphotographs of different bacterial strains co-cultured with organoids, but rows are not described. Are these examples of photographs taken? How many were there to calculate the bars in the lower graph. The graph B should use same labels as the microphotographs above. This would allow to present data without the “IG” – which as abbreviation is not determined in the manuscript.
Response 13: Thank you for your comments. The pictures shown are just some examples. Figure B shows statistics of fluorescence intensity of more than 10 samples. IG originally represent the organoids infected by bacteria, but it was not clearly marked before. In the revised manuscript, we replaced the abbreviation of IG with the letter of control.
Point 14: Fragment between lines 492 and 512 (with Fig. 9) – The short description of the results sound interesting, but there are no results shown. The figure 9, which should contain results from transcriptional analysis co-cultured organoids, does not supports the claims made in the text fragment, e.g. downregulation of TIRB3, ADM2, and PCK2 is nowhere to be seen. Moreover, the caption to figure 9 suggest that the illustration presents expression of Klebsiella and Clostridium and needs to be corrected. The figure itself is provided in low resolution and thus it makes reading out of it problematic. This whole fragment needs to be improved. I am not convinced by the presentation of results.
Response 14: Thank you for your comments. The lines of 496-500 should be deleted since it is repeated with the following paragraph (lines 504-510). In addition, I have attached the information related to gene expression data (Table S1).
Point 15: Line 537 “overseas” – where?
Response 15: Thank you for your correction. We have revised this misrepresentation.
Point 16: Generally, all figure cations need to be improved e.g. by introducing explanation of used abbreviations, labeling of data, etc.
Response 11: Thank you for your comments. All questions have been revised one by one.

Reviewer 2 Report
In current manuscript, ‘Effects of different feeding methods on the structure, metabolism, and gas production of infant and toddler intestinal flora and their mechanisms’, the authors have evaluated effects of different feeding methods on the characteristics of intestinal flora and gas production in infants and toddlers by using an in vitro simulated intestinal microecology fermentation and organoid model. However, the authors should clarify some major points .
- The authors should prepare suitable figures for the scientific paper. Some figures are not well organized. For example: the figures of this manuscript have wrong aspect ratios. Figure legends are not well placed and Figure captions are insufficient.
- Figure_1 and figure_8 A (have no information on Y axis) should be prepared again for better interpretation.
- Authors should mention the softwares’ name (in figure captions) utilized to draw the images of this manuscript.
- Authors have written in #line_623-624 that ‘Clinical trials have also shown that formula supplemented with 2′-FL resulted in plasma levels of inflammatory cytokines that were more similar to those of exclusively BF infants [52].’ The authors should explain the possible reason behind these findings.
- Authors have claimed that 2’-FL supplementations cannot be used by harmful bacteria and also it inhibit the colonization of harmful bacteria. How this selectivity towards harmful bacteria rather than beneficial ones?
- Fungi of fungal pathogens are also a part of human gut microflora. However, the authors have not presented any data regarding this aspect using their tested feeding methods.
- The important findings and novelty of this work should be mentioned in the ‘conclusion’ section.
Author Response
In current manuscript, ‘Effects of different feeding methods on the structure, metabolism, and gas production of infant and toddler intestinal flora and their mechanisms’, the authors have evaluated effects of different feeding methods on the characteristics of intestinal flora and gas production in infants and toddlers by using an in vitro simulated intestinal microecology fermentation and organoid model. However, the authors should clarify some major points.
Response: Thank you for your comments. All suggestions you pointed out have been revised and answered.
Point 1: The authors should prepare suitable figures for the scientific paper. Some figures are not well organized. For example: the figures of this manuscript have wrong aspect ratios. Figure legends are not well placed and Figure captions are insufficient.
Response 1: Thanks for your comments. We have redrawn some of the incorrect figures and modified the description.
Point 2: Figure_1 and figure_8 A (have no information on Y axis) should be prepared again for better interpretation.
Response 2: Thank you for your comments. We have modified the two figures.
Point 3: Authors should mention the software’s’ name (in figure captions) utilized to draw the images of this manuscript.
Response 3: Thank you for your suggestions. I have put relevant descriptions in the section of statistical analysis.
Point 4: Authors have written in #line_623-624 that ‘Clinical trials have also shown that formula supplemented with 2′-FL resulted in plasma levels of inflammatory cytokines that were more similar to those of exclusively BF infants [52].’ The authors should explain the possible reason behind these findings.
Response 4: Thank you for your comments. I think it may be related to the role of 2 '-FL in strengthening and supporting immune development and regulation.
Point 5: Authors have claimed that 2’-FL supplementations cannot be used by harmful bacteria and also it inhibit the colonization of harmful bacteria. How this selectivity towards harmful bacteria rather than beneficial ones?
Response 5: Thank you for your comments. Some previous studies have reported that 2 '-FL can promote the growth of some beneficial bacteria such as bifidobacteria, but there is the strain specificity. The inhibition of the 2 '-FL to harmful bacteria, I think it maybe by regulating the expression of intestinal epithelial pathogenic genes/signaling pathways, thus reducing the proliferation of the harmful bacteria in the gut.
Point 6: Fungi of fungal pathogens are also a part of human gut microflora. However, the authors have not presented any data regarding this aspect using their tested feeding methods.
Response 6: Thank you for your comments. The question you raised is very important, indeed, fungi are also a very important part of intestinal microecology. However, our study mainly focuses on the prokaryotic bacteria in the intestinal tract, and has not involved fungi for the time being, so it is worth exploring in the follow-up research.
Point 7: The important findings and novelty of this work should be mentioned in the ‘conclusion’ section.
Response 7: Thank you for your suggestions. I have added relevant expressions in the conclusion.
